# Analyzing Construction Workers’ Unsafe Behaviors in Hoisting Operations of Prefabricated Buildings Using HAZOP

**DOI:** 10.3390/ijerph192215275

**Published:** 2022-11-18

**Authors:** Lianbo Zhu, Hongxin Ma, Yilei Huang, Xun Liu, Xiaojin Xu, Zhenqun Shi

**Affiliations:** 1School of Civil Engineering, Suzhou University of Science and Technology, Suzhou 215000, China; 2Department of Construction Management, East Carolina University, Greenville, NC 27858, USA; 3Hangzhou Yongdu Real Estate Development Ltd., Hangzhou 311400, China

**Keywords:** unsafe behavior, prefabricated buildings, HAZOP

## Abstract

Along with the increasing number of prefabricated buildings being constructed in China each year, the incident rate of hoisting operations has been continuously rising. In order to improve construction safety in hoisting operations of prefabricated buildings, this paper analyzes the construction workers’ unsafe behaviors using the Hazard and Operability (HAZOP) method. A questionnaire survey and a literature review were first performed to gather information on safety risks and influencing factors during each stage of hoisting operations, and the survey results were statistically analyzed using the SPSS software. Next, HAZOP was applied to identify the deviation and change of the unsafe behaviors as well as their causes, consequences, and countermeasures. Finally, a case study was presented to verify the effectiveness of the countermeasures through a comparison and evaluation method from experimental economics. This paper demonstrates the use of HAZOP to analyze construction workers’ unsafe behaviors in hoisting operations of prefabricated buildings, and effective countermeasures in each stage of hoisting operations are proposed to mitigate unsafe behaviors. This paper therefore provides an innovative method and a theoretical foundation for reducing unsafe behaviors in hoisting operations of prefabricated buildings and serves as a reference for decision-making for hoisting safety policies in prefabricated construction projects.

## 1. Introduction

The number of prefabricated building projects has been growing rapidly in China under the strong support of recent policies. The Opinions on Promoting Green Development of Urban and Rural Construction press conference in October 2021 revealed that the gross area of new prefabricated construction projects in China reached 740 million square meters in 2021, an increase of 18% compared with 2020, accounting for roughly a quarter of all new construction project types; around two-thirds of the new prefabricated construction projects were concrete structures with a total gross area of 490 million square meters [1]. Along with this growth in prefabricated buildings, the incident rate of hoisting operations has been continuously rising. The Ministry of Housing and Urban-Rural Development of China defines a construction safety incident as an accident that occurs suddenly in the production and operation activities of the on-site construction unit, harming personal safety and health, damaging equipment and facilities, or causing economic losses, resulting in the temporary suspension or permanent termination of the original production and operation activities. The Ministry’s 2021 Housing and Municipal Engineering Production Safety Accidents Report, as shown in Table 1, reveals that 99 safety incidents and 123 deaths occurred during the first quarter of 2021, an increase of 17 incidents (21%) and 22 deaths (22%) compared with the same period of the previous year. Among the fatalities, 54 were falls and 12 were from crane injuries.

Because most of these fatalities occurred in the construction and hoisting stage and have posed a serious threat to the safety of workers, it is of significant importance that construction workers’ unsafe behaviors in hoisting operations of prefabricated buildings are identified and analyzed. To achieve this goal, this study uses the Hazard and Operability (HAZOP) method combined with survey questionnaires, statistical analysis, and field verification through a case study to improve construction safety in the hoisting operations of prefabricated buildings.

## 2. Literature Review

Safety behavior refers to behavior that will not infringe, endanger, harm, or cause los, and which is affected by personality traits [2], emotion control [3], work–life conflicts [4,5], etc. The unsafe behaviors of employees can cause many injury incidents, especially in the construction industry. At present, research on the unsafe behavior of construction workers mainly focuses on the safety culture at the group level and behavioral safety at the individual level. For example, He [6] discussed the differences between supervisors and workers in the dimensions of safety environment and safety behavior at the group level, while Zhang [7] started from the perspective of the individual level. The paper adopts a psychological method to study how personality traits affect the occurrence of unsafe behaviors of construction workers.

Quantitative analysis methods are widely used in the study of workers’ unsafe behavior. Guo [8] used the skeleton-based real-time identification method, developed by Guo to capture the dynamic movement of workers, to identify unsafe behaviors. Qi [9] revealed the occurrence mechanism of workers’ unsafe behaviors from the perspective of occupational stress through the developed structural equation model. Hung [10] used a deep convolutional neural network to propose a system to identify workers’ dangerous behaviors and classify unsafe behaviors. Qualitative analysis methods have also been used in studying workers’ unsafe behaviors and have focused on areas such as social psychology and modular construction. Khan [11] evaluated fire safety performance according to the characteristics of multi-story garment (RMG) buildings in India, and Zermane [12] used fault tree analysis to assess the risk of high-altitude work in Malaysian buildings. Chih [13] studied the negative impact of psychological contract breach (PCB) on workers’ safety, Guo [14] used an intelligent video surveillance method to carry out an early warning of workers’ illegal behaviors, and Chatzimichailidou [15] studied the safety risk of modular construction in the construction industry. Although these studies have revealed the causes of workers’ unsafe behaviors to a certain extent, they are limited to traditional buildings and rarely involve new trends such as prefabricated buildings.

The research on occupational safety in the field of prefabricated construction mainly focuses on safety-influencing factors and intervention measures. James [16] proposed that the standardized use of mechanical equipment can effectively reduce the occurrence of safety accidents in prefabricated construction. Lee [17] used the AHP method to analyze the weight of the influencing index of safety management in prefabricated construction and evaluated project safety based on it. Francisco [18] put forward a risk assessment theory applicable to the construction site of prefabricated buildings, which concluded that safety accidents due to falling from high altitude and lifting injury accounted for a high proportion and suggested that the causes of accidents in the process of installation and demolition were due to non-compliance with operating procedures and other unsafe behaviors. Fard [19] made statistics on the data of safety accidents of prefabricated buildings in the United States in recent years and concluded that in falls from height, instability of the connection points of prefabricated members was the main factor inducing unsafe behaviors. In a study of intervention measures, Zou [20] focused on the relationship between prefabricated construction personnel and the management system and concluded that the key to safety management for prefabricated construction lies in human management. Xu [21] and Zhang [22] identified and analyzed the unsafe behaviors of construction workers using computer vision and proposed targeted prevention and treatment measures. Chang [23] reduced the construction safety risk level of prefabricated buildings to the maximum extent through a novel dual-objective optimization model. However, most of the above studies focused on the whole process of prefabricated building construction: the analysis was not detailed enough due to the large scope of involvement. In addition, more attention was paid to the hazard factors leading to unsafe behaviors, while less attention was paid to the possible consequences of unsafe behaviors and subsequent preventive measures.

The main research fields for HAZOP analysis are concentrated in petrochemical, thermal, and chemical engineering. Marhavilas [24] used HAZOP–DMRA technology to evaluate the safety of petrochemical industry operations, while Sirima [25] chose fault tree analysis and HAZOP analysis to explore the risks in the steam production process. Patle [26] proposed a holistic method for the safety analysis of the formic acid production process by using the improved HAZOP model. HAZOP analysis is often used in combination with sensitivity analysis and survey questionnaires. Sbaaei [27] used a sequential module simulation method and automated HAZOP study to conduct dynamic modeling. Zhao [28] used the deep learning method to improve the safety assessment process of HAZOP in the chemical production process. Danko [29] and Yang [30] both used survey questionnaires to simulate the “expert brainstorming” segment of HAZOP analysis. Some scholars also applied HAZOP analysis to the field of construction; for example, Niu [31] used HAZOP analysis to identify and manage the unsafe behavior of crane workers, and Wang [32] combined this method with accident analysis to build a knowledge database of construction expert systems.

Although various studies have investigated the unsafe behaviors of construction workers, few have focused on the unsafe behaviors during the construction and hoisting operation of prefabricated buildings. Since the HAZOP analysis method can be utilized to identify the deviation and change of unsafe behaviors as well as their causes, consequences, and countermeasures, it was applied in this study to investigate the unsafe behaviors during hoisting operations of prefabricated buildings. The application of HAZOP provides an innovative method and theoretical foundation for reducing unsafe behaviors in hoisting operations of prefabricated buildings and serves as a reference for decision-making for hoisting safety policies in prefabricated construction projects.

## 3. Methodology

Based on the findings of the literature review, a survey questionnaire was first developed and administered at selected construction sites of prefabricated buildings to gather respondents’ perceptions of potential safety risks and influencing factors. The collected survey dataset was then statistically analyzed using the SPSS software to ensure its integrity and reliability. Next, the survey results were summarized to identify the safety risks and safety-influencing factors during each stage of hoisting operations. After that, HAZOP was applied to identify the deviation and change of the unsafe behaviors as well as their causes, consequences, and countermeasures. Finally, a case study was presented to verify the effectiveness of the countermeasures through comparison and evaluation method in experimental economics.

### 3.1. Survey Questionnaire

The survey was administered at a construction site of prefabricated buildings located in the city of Zibo, China to ensure the sample size and representativeness. A total of 300 survey questionnaires were distributed on-site to hoisting workers, of which 263 were recovered (87.7%). After sorting out the incomplete questionnaires, 203 were determined to be valid (67.7%).

The questionnaires are divided into four sections with objective questions in each section, including (1) basic personal information such as gender, age, occupation, etc., (2) workers’ perception of risks during hoisting operations, (3) the most common safety hazards during hoisting operations, and (4) the causes of safety hazards during hoisting operations. The workers’ perception of risks during hoisting operations describes the degree of danger perceived in different phases of hoisting operations. The most common safety hazards during hoisting operations include a list of 33 potential safety hazards concluded from the literature review based on a 5-point Likert scale. The causes of safety hazards include a list of 16 potential factors contributing to the occurrence of safety hazards based on a 5-point Likert scale.

### 3.2. HAZOP Analysis

Since the HAZOP analysis method can be utilized to identify the deviation and change of unsafe behaviors as well as their causes, consequences, and countermeasures, it was applied in this study to investigate the unsafe behaviors during hoisting operations of prefabricated buildings. The lead words of the HAZOP analysis were first created to define the workers’ behaviors and describe the various conditions that can occur during hoisting operations. Table 2 presents the lead words and their meaning used in the HAZOP analysis, which guide the identification of deviations of safety factors in hoisting operations.

### 3.3. Case Study

A residential prefabricated building project was selected as the case study for the HAZOP analysis. The case study uses the comparison and evaluation method in experimental economics, which is able to obtain the density relationship between the evaluation results and the probability distribution after applying independent variables, allowing a clearer view of the impact of independent variables on the evaluation results. As a result, the comparison and evaluation method can be used to verify the validity of the hypothesis.

## 4. Results

### 4.1. Survey Statistics

The analysis of valid questionnaires showed that male workers accounted for 89.4%, while female workers accounted for 10.6%, indicating a typical male-dominated workforce in the construction industry. Table 3 illustrated the distribution of age and years of work experience of the surveyed construction workers. Construction workers over 50 years old accounted for the largest proportion (37%) of the surveyed population, suggesting a serious aging trend among construction workers in China. While construction workers with 1–3 years of work experience account for the largest proportion, those with over 5 years of work experience account for the smallest proportion, indicating a lack of experienced workers in the field of prefabricated construction and that many workers have transitioned from traditional fields of the construction industry in recent years.

Next, the statistical software package SPSS 26.0 was used to analyze the reliability of the questionnaire survey results, as shown in Table 4. The number of valid observations included in the analysis was 203 (77.2%), and the missing value was 60 (22.8%). The calculation result of Cronbach α coefficient α = 0.986 indicates good reliability of the survey data.

### 4.2. Safety Risks

The hoisting operations in prefabricated buildings were divided into six phases according to the typical prefabricated construction sequence, which include component loading and unloading, stacking, lifting, installation, positioning, and correction. Out of the 33 potential safety hazards concluded from the literature review, 23 were determined to be high-risk safety hazards from the survey results and reliability analysis with a coefficient above the average score of 4.62 based on a 5-point Likert scale. These 23 safety risks are detailed in Table 5 and categorized into each of the six phases of hoisting operations.

### 4.3. Safety Influencing Factors

After analyzing the survey results, 13 safety factors affecting the hoisting operations of prefabricated buildings were determined to be significant among the 16 potential factors with a coefficient above the average score of 4.58 based on a 5-point Likert scale. In addition, these safety factors were classified into four safety evaluation indicator categories, namely human, material, environmental, and managerial, based on the research findings of Bao [33], Wang [34], and Singh [35]. The influencing factor of each safety factor is identified and summarized in Table 6.

### 4.4. Unsafe Behaviors

HAZOP analyses were performed for each of the six phases of hoisting operations in prefabricated buildings identified in Table 5 using the selected lead words in Table 2. The results of HAZOP analysis for each phase present the deviation, reasons, consequences, and countermeasures for construction workers’ unsafe behaviors in hoisting operations of prefabricated buildings. Table 7 shows the HAZOP analysis results for the loading and unloading phase, where countermeasures are proposed to strengthen the inspection and maintenance of equipment and improve the safety risk protection skills of hoisting workers.

Table 8 shows the HAZOP analysis results for the stacking phase, where the proposed countermeasures include strengthening regular inspection and maintenance, hanging code warning signs, isolating the stacking site, and implementing the reward and punishment mechanism.

Table 9 presents the HAZOP analysis results for the lifting phase, where the countermeasures are proposed to improve and implement a comprehensive acceptance system for components entering the site, rationally arrange the lifting points of components, pay attention to warnings during lifting operations, and install a torque limiter to prevent the crane from overturning.

Table 10 presents the HAZOP analysis results for the installation phase, where the countermeasures are proposed to carry out special drill activities for construction safety, implement safety protection measures at the operation site (such as safety fences, double hanging points to secure exterior wall scaffold, etc.), verify the construction plan, and assign special personnel for power management.

Table 11 presents the HAZOP analysis results for the positioning phase, where the countermeasures are proposed to emphasize the importance of worker safety and protection, strengthen on-site safety inspection, and check whether the operation meets the operating standards.

Table 12 shows the HAZOP analysis results for the correction phase, where the proposed countermeasures include strengthening on-site inspection and template connection.

### 4.5. Countermeasures

Based on the HAZOP analysis results, the significant unsafe behaviors in the six phases of hoisting operation in prefabricated buildings can be summarized as follows: (1) in the component loading and unloading phase, it is easy to hurt workers; (2) in the stacking phase, component collapse easily occurs; (3) in the lifting phase, components easily fall or cause injury due to illegal equipment operation; (4) in the installation phase, workers are easily injured by components or due to the lack of safety protection at the operation site; (5) in the positioning phase, collapse of cast-in-place structure easily occurs; (6) in the correction phase, formwork collapse or mold-moving incidents easily occur. Based on these significant unsafe behaviors determined by the HAZOP analyses, the following prevention and control countermeasures are proposed:Countermeasures to prevent workers from being injured in the component loading and unloading phase:
Strengthen the technical training of operation personnel and improve the safety risk protection skills of hoisting workers.Improve and implement a comprehensive acceptance system of component entry.Strengthen the regular inspection and maintenance of loading and unloading equipment such as component transport frames and the hoisting crane.Countermeasures to prevent the collapse of components in the stacking phase:
Improve and implement a safety disclosure system, a reward and punishment mechanism, and a comprehensive safety control assessment system, and prohibit the occurrence of too many layers of prefabricated components or missing pads.Hang safety warning signs at the component stacking site, and take moisture-proofing and rainproofing.Improve the security and stability of the component storage frame.Replace the component storage rack with potential safety hazards in time.Countermeasures to prevent falling or injury incidents caused by illegal equipment operation in the component lifting phase:
Calculate the maximum lifting mass and use it as a reference to select the appropriate equipment model.Rationalize the arrangement of hanging points to meet requirements.Use red flags to temporarily isolate the affected area of lifting operation from other operating areas.Ensure that the weight balance bar and guide rope are reasonably configured in the process of lifting to avoid the flipping phenomenon caused by wind factors.Clean up foreign matter in the lifting point before lifting the member, and check whether the lifting clip pin and the bending ring are closely connected.Install a torque limiter to prevent the crane from overturning.Countermeasures to prevent component injuries to workers and worker casualties caused by a lack of safety protection at the operation site during the component installation phase:
Use double hanging points to secure exterior wall scaffolding.Ensure that hoisted workers hook the steel bar of the column head with the self-locking safety belt through the core when installing prefabricated columns, walls, and plates.Carry out special drills for construction safety periodically.Give priority to the installation of the main safety cable when the main beam is hoisted.Ensure that the height of the floor enclosure is strictly greater than 1.8 m.Ensure that, for electricity use, cable lines are laid in accordance with regulations.Countermeasures to prevent cast-in-place structure collapse during the component positioning phase:
Strengthen the on-site inspection of cast-in-place concrete strength.Put up safety signs.Implement a punishment system for unsafe behaviors.Countermeasures to prevent template collapse or mold-moving incidents during the component correction phase:
Define the scope of responsibility and work content for departments or individuals through the system in detail, and ensure that responsibilities are met.Ensure a good template connection and reinforcement support.Carry out regular inspection of on-site operation safety to eliminate potential safety hazards.

The proposed countermeasures for workers’ unsafe behaviors in the six stages of hoisting operations are further summarized into four categories based on the classification of safety-evaluation indicators, as shown in Table 13.

## 5. Field Verification

### 5.1. Experiment Design

The residential project selected for the case study consists of 14 separate residential buildings numbered Building 1 through Building 14, each containing eleven levels above ground and two levels underground, with a total height of 33.35 m and an elevation difference of 0.45 m between the interior and exterior floor height. The building structure was rated at safety grade two, seismic fortification intensity seven degrees, building site category three, and building anti-floating design grade b.

One independent variable was selected in the case study experiment in order to avoid the uncertain influence of variable mixing and crossing in the experiment, which was whether to implement HAZOP analysis for unsafe behaviors and countermeasures. The countermeasures for workers’ unsafe behaviors in hoisting operations of prefabricated buildings were obtained from the results of the HAZOP analyses presented in this paper.

Two hoisting construction teams in this project were marked as Team A and Team B, and both teams had 30 hoisting workers after adjustment. Team A was selected as the experimental group and strictly implemented the HAZOP analysis for unsafe behaviors and countermeasures in six phases of hoisting operations, whereas Team B was selected as the control group, which maintained its original working habits and did not implement any of the countermeasures for the unsafe behaviors. The duration of the case study was 20 days, which was roughly the schedule to complete the main structure of the four-story building. Due to the maintenance of the structural concrete of each floor, hoisting operations were performed alternatively between two buildings for both teams.

All hoisting workers were asked to wear a wristband sensor to detect when and what unsafe behaviors happened. The jobsite was equipped with 24-h surveillance cameras to capture the type and severity of hoisting workers’ unsafe behaviors. Drone footage was also used to analyze safety risks due to negligence. Through these monitoring measures, the probability density between the independent variable and a reduction in unsafe behaviors was calculated, which consequently determined whether the countermeasures from the HAZOP analysis did help reduce the occurrence of hoisting workers’ unsafe behaviors.

### 5.2. Experiment Results

The actual experiment case study lasted for 22 days from 29 July to 19 August 2021, including two days of delay due to typhoon weather. Team A completed four floors, including Building 3 floors 2 and 3 and Building 5 floors 2 and 3, while Team B also completed four floors, including Building 1 floors 3 and 4 and Building 6 floors 4 and 5.

By analyzing the wristband sensor data, on-site surveillance images, and drone footage, the unsafe behaviors of the two teams were compared with the safety deviation obtained by the HAZOP analysis method above. After removing the unsafe events that were irrelevant to this study, such as worker heatstroke, fighting, etc., three unsafe behaviors were found in Team A and nine unsafe behaviors were identified in Team B, as shown in Table 14.

A comparison of worker unsafe behaviors between the two hoisting teams in the case study experiment demonstrates that the HAZOP analysis method can be effectively applied to the study of construction workers’ unsafe behaviors and that the countermeasures obtained by the HAZOP analysis method can significantly reduce the occurrence of unsafe behaviors in prefabricated building hoisting operations.

## 6. Conclusions

Safety management of hoisting operations in prefabricated construction plays a significant role in the overall construction safety of prefabricated building projects. Based on the construction process, hoisting operations in prefabricated buildings were divided into six separate phases in this paper, and the HAZOP method was utilized to systematically analyze workers’ unsafe behaviors. Through survey questionnaires, 23 unsafe behaviors of hoisting workers were identified during the six hoisting phases, and 13 safety influencing factors were determined through a comprehensive literature review. The results of HAZOP analyses revealed significant unsafe behaviors in each of the six phases of hoisting operation in prefabricated buildings and proposed respective countermeasures. In addition, the results of HAZOP analyses were verified by using a comparison and evaluation method from experimental economics through a case study experiment. The findings of this paper provide a theoretical foundation for the analysis and research of construction workers’ unsafe behaviors in hoisting operations and have significant importance in reducing the incident rate in prefabricated building construction. This paper also serves as a reference for decision-making for hoisting safety policies in prefabricated construction projects.

## Figures and Tables

**Table 1 ijerph-19-15275-t001:** Comparison of statistics on construction safety incidents in China.

Construction Incident Type	Q1 2021	Q1 2020
Construction safety incidents	99	82
Death toll	123	101
From falling incidents	54	-
From lifting injury incidents	12	-

**Table 2 ijerph-19-15275-t002:** Lead words and meaning of the HAZOP analysis.

Lead Words	Meanings
none	The established workflow is not followed at all
more	Excessive working hours or intensity
less	Some protective measures are missing
as well as	There are unnecessary actions to achieve the goal
part of	Only part of the goal is achieved
reverse	The operation process is completely different from the requirements
other than	Events that do not meet the design requirements occur

**Table 3 ijerph-19-15275-t003:** Distribution of age and years of work experience of surveyed construction workers.

Age			Years of Work Experience		
20–30	35	17%	<1 years	31	15%
31–40	61	30%	1–3 years	89	44%
41–50	33	16%	3–5 years	73	36%
>50	74	37%	>5 years	10	5%

**Table 4 ijerph-19-15275-t004:** Reliability statistics of survey data.

Reliability Statistics		
Cronbach’s Alpha	Cronbach’s Alpha Based on Standardized Items	N of Items
0.986	0.987	49

**Table 5 ijerph-19-15275-t005:** Identification of safety risks in hoisting operations of prefabricated buildings.

Hoisting Operation Phase	Safety Risks
Component loading and unloading	The special iron frame for transportation is not fixed
	Component imbalances
Component stacking	The stiffness of the member storage frame is poor
	Incorrect position of wooden support
	Obstacles in the stacking area
	Too many stacked prefabricated floors (>6)
Component lifting	The embedded lifting ring of the component falls off
	The lifting point deviates from the center of gravity of the component
	Workers stay within the moving range of mechanical boom
	Overload operation of lifting equipment
	Crane overturning and lifting parts falling
Component installation	The lowering speed of components is too fast, or the components are not kept horizontal when falling
	Collapse of exterior wall scaffold
	The lower hanging point of the member is not connected by a hoist
	No guardrail is installed in elevator shaft and other places
	Embedded rebar trips workers
	High-altitude-edge working equipment or personnel fall
	Neglecting electrical safety
Component positioning	Prefabricated overlap plate-positioning correction squeezing hands
	Transverse collision of precast outer wall
	The strength of concrete does not meet the design requirements when removing the limiter
Component correction	Collapse of prefabricated composite plate support
	Precast wall panels collapse

**Table 6 ijerph-19-15275-t006:** Safety-influencing factors in hoisting operations of prefabricated buildings.

Category	Safety Factors	Influencing Factors
Human	Technical level of employees [36]	Hoisting workers’ technical mastery and technical proficiency.
	Employee collaboration literacy	The ability of workers to cooperate with each other in construction.
	Safety awareness level of construction personnel [37]	Safety awareness and habits of site operators.
Material	Quality of assembled components	Prefabricated part factory quality or whether damage occurs during transportation.
	Hoisting machine selection	Whether the maximum lifting mass of the selected lifting equipment is lower than the maximum mass of the components.
	Stability of support system [35]	Whether scaffold and support system are stable.
	Protection degree of field operation	Whether there is protection around the foundation pit, whether there is a protection net, guardrail, etc.
Environmental	Severe natural weather	Whether workers operate in fog, typhoon, snowstorm, and similar environments.
	Site safety atmosphere [36]	Whether the site hangs safety banners, whether the construction team emphasizes safety risks.
Managerial	Safety training	Whether lifting worker safety training is organized on a regular basis.
	Safety management system	Whether there is a perfect and reasonable safety management system.
	On-site safety supervision inspection	Whether the construction site supervisor often carries out inspections.
	Construction scheme planning	Whether the management keeps the operators fully informed of the construction scheme and planning.

**Table 7 ijerph-19-15275-t007:** HAZOP analysis of the components loading and unloading phase.

Lead Word	Deviation	Reason	Consequence	Countermeasure
none	The special iron frame for transportation is not fixed	The stability of the support system is poor	Prefabricated parts hurting workers	Regular inspection and maintenance of special transport frame
reverse	Component imbalances	The technical level of employees is low	Prefabricated parts hurting workers	Hoisting point layout rationalization design

**Table 8 ijerph-19-15275-t008:** HAZOP analysis of the components stacking phase.

Lead Word	Deviation	Reason	Consequence	Countermeasure
other than	The stiffness of the member storage frame is poor	The stability of the support system is poor	Prefabricated parts hurting workers	Check and maintain component storage regularly
reverse	Incorrect position of wooden support	Low level of safety awareness among workers	Member collision with workers when lifting	Warning board for specifications of suspension skids
as well as	Obstacles in the stacking area	No hidden danger investigation was carried out at the stacking site	Worker injury due to crane collision with an obstacle	Take isolation measures at the stacking site
more	Too many stacked prefabricated floors (>6)	Lack of safety operation training	Workers injury due to prefabricated floors collapse	Establish and implement a corresponding reward and punishment mechanism

**Table 9 ijerph-19-15275-t009:** HAZOP analysis of the components lifting phase.

Lead Word	Deviation	Reason	Consequence	Countermeasure
part of	The embedded lifting ring of the component falls off	The quality of assembled parts is poor	A member falls from a high altitude and injures workers	Improve and implement a comprehensive acceptance system for the entry of components
reverse	The lifting point deviates from the center of gravity of the component	Practitioners have low cooperation literacy	A member falls from a high altitude and injures workers	Introduce a hoisting point layout rationalization design
other than	Workers stay within the moving range of mechanical boom	The level of safety awareness of construction personnel is low	Mechanical boom collides with workers	Flag the working area is with red pennants
more	Overload operation of lifting equipment	Wrong hoisting machine selection	Equipment failure resulted in worker injury	Calculate the maximum lifting mass and select the equipment model based on it
other than	Crane overturning and lifting parts falling	Severe natural weather	Worker casualties and economic losses	Add a moment limiter and conduct qualified acceptance for crane support foundation

**Table 10 ijerph-19-15275-t010:** HAZOP analysis of the component installation phase.

Lead Word	Deviation	Reason	Consequence	Countermeasure
more	The lowering speed of components is too fast, or the components are not kept horizontal when falling	The level of safety awareness of construction personnel is low	Structural injuries to workers	Periodically carry out construction safety drills
as well as	Collapse of external wall scaffold	The supervision of hanging point installation is not strict	Worker death by high-altitude fall	Adopt an installation mode of hanging points and double fuses
none	The lower hanging point of the member is not connected by a hoist	The technical level of employees is low	Worker injury due to component slippage during the lifting process	Have the construction plan verified by experts
none	No guardrail is installed in elevator shaft and other places	The protection degree of field operation is weak	Worker falls due to stepping into the air	Set a safety barrier with a height greater than 1.8 m
as well as	Embedded rebar trips workers	The protection degree of field operation is poor	Worker injury	Label and cover embedded parts with film
as well as	High-altitude-edge working equipment or personnel fall	The protection degree of field operation is weak	Lift workers fall to death or injure workers below	Install safety barriers and wear self-locking belts
less	Neglecting electrical safety	On-site safety supervision inspection is not strict	Worker death by electrocution	Restrict operation and work to specially assigned and certificated personnel

**Table 11 ijerph-19-15275-t011:** HAZOP analysis of the components positioning phase.

Lead Word	Deviation	Reason	Consequence	Countermeasure
as well as	Prefabricated overlap plate-positioning correction squeezing hands	Poor safe operation training	Worker injury	Equip workers with protective equipment
other than	Transverse collision of precast outer wall	Practitioners have low cooperation literacy	Worker injury or death by falling	Check to ensure that the angle between the sling and the plate plane is between 45° and 60°
less	The strength of concrete does not meet the design requirements when removing the limiter	Construction scheme planning is not comprehensive	Collapse of concrete causes worker injury or death	Strengthen on-site supervision and inspection

**Table 12 ijerph-19-15275-t012:** HAZOP analysis of the component correction phase.

Lead Word	Deviation	Reason	Consequence	Countermeasure
part of	Collapse of prefabricated composite plate support	The stability of the support system is poor	Worker death and injury	Harden template connections
part of	Precast wall panel collapse	The stability of the support system is poor	Worker death and injury	Use wall braces and seven-letter codes for reinforcement

**Table 13 ijerph-19-15275-t013:** A summary of countermeasures for workers’ unsafe behaviors in hoisting operations.

Category	Countermeasure
Human	Strengthen the technical training and assessment of operators
Material	Strengthen regular inspection and maintenance of equipment
Environmental	Improve and implement rules and regulations for operation in severe weather conditions
	Strengthen the safe working atmosphere of the construction team
Managerial	Improve and implement various safety management systems
	Carry out special drills for construction safety periodically
	Implement regular inspection of on-site operation safety

**Table 14 ijerph-19-15275-t014:** Comparison of workers’ unsafe behaviors between the two hoisting teams.

	No.	Date	Location	Description
Team A	1	4 August	Building 3 Floor 2	Portions of scaffolding safety net are not fully secured.
	2	4 August	Building 5 Floor 2	The outer wall component fell off during lifting.
	3	17 August		A worker squeezed his finger while positioning and correcting the laminated plates.
Team B	1	2 August		A worker squeezed his finger while positioning and correcting the laminated plates.
	2	2 August	Building 6	The third batch of prefabricated laminated slabs were stacked too high.
	3	5 August	Building 1	The windowed wall members of the kitchen were not stacked correctly with special storage racks.
	4	11 August	Building 6 Floor 4	The outer wall component fell off during lifting.
	5	13 August	Building 1 Floor 4	The stair safety guardrail shelter is not complete
	6	13 August	Building 1 Floor 4	The elevator shaft protection shelter is not complete.
	7	13 August		Overturning of concrete pump truck
	8	16 August	Building 1	Multiple machines used the same circuit breaker.
	9	17 August	Building 6 Floor 5	The mold moved when the floor was being poured.

## Data Availability

The data are available from the corresponding author upon request.

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
