# Peer review of "Analyzing Construction Workers’ Unsafe Behaviors in Hoisting Operations of Prefabricated Buildings Using HAZOP"

_ijerph, 2022, doi:10.3390/ijerph192215275_

Round 1

Reviewer 1 Report (Previous Reviewer 1)

Τhe submitted manuscript has taken into consideration the majority of my comments in its revised form, and is more coherent, well structured and scientifically sound, thus my proposal is to be published after the following minor corrections:  

-      The phrase “The findings…behaviors” in lines 21-24 of the abstract must be revised in better syntax.

-      In subsection 3.1, the development of the questionnaire and its content validation should be briefly presented.

-      In Table 2, the definition of lead word “as well as” must be corrected.

-      Table 3 is not necessary, since these data are already given in the text.

Author Response

Dear Editor,

We much appreciate the reviewers’ time and suggestions on improving the quality of this paper. We have thoroughly addressed each of the reviewer’s comments in the table below along with the revised manuscript with Track Changes turned on.

Reviewer 1’s Comment

Authors’ Response

The phrase “The findings…behaviors” in lines 21-24 of the abstract must be revised in better syntax.

The sentence has been revised to “This study demonstrated the approach of using HAZOP to analyze construction workers’ unsafe behaviors in hoisting operations of prefabricated buildings, and effective countermeasures in each stage of hoisting operations were proposed to mitigate unsafe behaviors.”

In subsection 3.1, the development of the questionnaire and its content validation should be briefly presented.

The following sentences have been added to explain the development of the questionnaire: “The workers’ perception of risks during hoisting operations describes the degree of danger perceived in different phases of hoisting operations. The most common safety hazards during hoisting operations include a list of 33 potential safety hazards concluded from literature review based on a 5-point Likert scale. The causes of safety hazards include a list of 16 potential factors contributing to the occurrence of safety hazards based on a 5-point Likert scale.”

The following sentences have been added to explain the validation of the questionnaire: “After sorting out the incomplete questionnaires, 203 were determined to be valid (67.7%).”

The following sentences have been added in section 4.2. Safety Risks to explain the results of the survey: “Out of the 33 potential safety hazards concluded from the literature review, 23 were determined to be high-risk safety hazards from the survey results and reliability analysis with a coefficient above the average score of 4.62 based on a 5-point Likert scale.”

The following sentences have been added in section 4.3. Safety Influencing Factors to explain the results of the survey: “After analyzing the survey results, 13 safety factors affecting the hoisting operations of prefabricated buildings were determined to be significant among the 16 potential factors with a coefficient above the average score of 4.58 based on a 5-point Likert scale.”

在表2中,必须更正引导词“以及”的定义。

该定义已修订为“为实现目标而采取不必要的行动”

表3不是必需的,因为这些数据已经在文中给出了。

表 3 已删除。

有关详细信息,请参阅附件。

Reviewer 2 Report (Previous Reviewer 2)

Dear Authors,

Overall, addressing the comments was very clear.

The heading structure requires more attention as there is similar headings (e.g., 3.3 Case Study, 5 Case Study)

The figures (1-2) can be put in one combined table. Tables (need 7-13) need to be same formatting to tables 1-5.

Kindest regards,

Evan

Author Response

Dear Editor,

We much appreciate the reviewers’ time and suggestions on improving the quality of this paper. We have thoroughly addressed each of the reviewer’s comments in the table below along with the revised manuscript with Track Changes turned on.

Reviewer 2’s Comment

Authors’ Response

The heading structure requires more attention as there is similar headings (e.g., 3.3 Case Study, 5 Case Study)

Section 5 has been renamed to “Field Verification”.

The figures (1-2) can be put in one combined table.

Figures 1 and 2 have been combined into a new Table 3.

表格(需要 7-13 个)的格式必须与表格 1-5 的格式相同。

表7-12和表14中的垂直分隔线已被删除,以便与其他表保持一致。

有关详细信息,请参阅附件。

This manuscript is a resubmission of an earlier submission. The following is a list of the peer review reports and author responses from that submission.

Round 1

Reviewer 1 Report

The topic of this manuscript (unsafe behaviors of construction workers) is within the scope of International Journal of Environmental Research and Public Health, and the intent to approach it with the implementation of a HAZOP analysis does create expectations for an innovative research.

However, the paper presents three different projects (a questionnaire survey, a HAZOP analysis, a case study) which, although they are presented as being related to the identification of unsafe behaviors in a specific work context (hoisting operations of prefabricated buildings), they are mostly related to job hazard analysis or risk assessment and mitigation.

Perhaps this is due to a misunderstanding of what is an unsafe behavior, and how can it be analysed as a deviation of a safe behavior, thus implementing a systematic and thorough analytical method, as HAZOP. Your reference #28 gives a very analytical example of this type of analysis.

While I acknowledge the field work done for this research project, such as surveys and on-site observations, the theoretical background, scientific methodology and analysis is very poor in order for the manuscript to be published, even with major revisions.

For this reason, my proposal is that the submitted manuscript should be rejected in its present form, and in case the authors want to revisit their work and resubmit, I offer them hereby some more specific comments:

1.    Introduction: The presentation of the broad context of the study and its importance is weak. More specifically:

-    Lines 37-39 (also in abstract). The incidents mentioned here are not known to the readers, therefore a further elaboration is needed, or this phrase should be omitted

-    The term “production safety incidents” is not commonly used in safety science, and since the paper is about construction safety a clarification should be given

-    Lines 39-44. The incident data referred to here are not clear. Do the numbers given refer to absolute numbers of construction accidents and fatalities or to incident rates (in which case a definition of how these rates are calculated should be given). Reference [2] is not available to the reviewer (perhaps it is in Chinese?), therefore a short table of incident data could be included.

-    Furthermore, the argument express in lines 44-47 about the significance of construction workers’ unsafe behaviors in hoisting operations of prefabricated buildings should be better supported, with more detailed data.

2.    Literature Review

-    If the paper continues to refer to “unsafe behaviors”, there is an extent literature on “behavior based safety” and it should be mentioned in this section. However, the authors are invited to consider other analytical approaches to their research.

-    The term “industrial workers in the construction field” is not commonly used in safety science, it would be better replaced by the term “construction workers”.

-    Editing is needed for English language/syntax and more formal style (i.e. line 65…in the research of this kind of problem, line 69…and other issues are generally high attention)

3.    Methodology

-    This section should present the methodology implemented in all three parts of the research (survey, analysis, case study), the selection of methods and tools should be adequately supported and their interrelation (i.e. subsequent steps?) should be discussed.

-    Information should be given on the development and validation of the questionnaire, as well as on the selection criteria and representativeness of the sample.

-    The section includes results of the analysis of the questionnaires (which should be part of a different Result section), however it is not clear whether these results all refer to respondents’ personal perceptions of safety risks and safety influencing factors, especially in par.3.3 (a correction in numbering in line 164 is needed) where results are combined (?) with existing literature research.

4.    HAZOP analysis of Unsafe Behaviors

-    HAZOP is usually implemented in order to identify hazards in complex systems, often in process industries. In this case, hazards have already been identified by the survey, while the reasons and the consequences are very general and self evident (i.e.” the training is poor”, “the safety management system is not perfect”, “prefabricated parts hurt workers” etc.) or in some cases questionable (as in the last item of Table 8). As a result, the majority of the proposed countermeasures are also very generic , related mainly to basic safety rules and safety management issues. Moreover, the “unsafe behaviors” presented in the first paragraph of section 4.3 as resulting from the HAZOP analysis, are mainly safety risks identified during the specific operations.

5.         Case study of HAZOP analysis

-       The “unsafe behaviors”, identified in the previous section, were “observed” (through wristbands, surveillance cameras and drones!) in two groups of prefabricated buildings’ workers and “HAZOP analysis was implemented in one of the groups”, thus resulted in lesser number of incidents. If this is what this section supports, I would expect the authors to explain what are the specific unsafe behaviors and how they were observed by these methods, not common in Behavior Based Safety Observations. Moreover how were the proposed countermeasures introduced and implemented, and what other parts of the HAZOP analysis were implemented in the experimental group.

-       The presentation of the methodology in lines 302-307 needs further elaboration and related references.

-       The conclusion in lines 343-346 is not adequately supported by the information given.

Author Response

请参考附件

Reviewer 2 Report

Dear Authors,

I'm really excited to see your research and investigations in the Prefabricated Building/works. I've some comments to improve your valuable article.

Kindest regards,

Author Response

请参考附件
